# Classification of Biodegradable Substances Using Balanced Random Trees and Boosted C5.0 Decision Trees

**DOI:** 10.3390/ijerph17249322

**Published:** 2020-12-13

**Authors:** Alaa M. Elsayad, Ahmed M. Nassef, Mujahed Al-Dhaifallah, Khaled A. Elsayad

**Affiliations:** 1Department of Electrical Engineering, College of Engineering, Prince Sattam Bin Abdulaziz University, P.O. Box 54, Wadi Aldawaser 11991, Saudi Arabia; a.nasef@psau.edu.sa; 2Computers and Systems Department, Electronics Research Institute, Giza 12622, Egypt; 3Department of Computers and Automatic Control Engineering, Faculty of Engineering, Tanta University, Tanta 31733, Egypt; 4Systems Engineering Department, King Fahd University of Petroleum & Minerals, Dhahran 31261, Saudi Arabia; mujahed@kfupm.edu.sa; 5Pharmacy Department, Cairo University Hospitals, Cairo University, Cairo 11662, Egypt; khaled.al.elsayad@std.pharma.cu.edu.eg

**Keywords:** QSAR, biodegradable substances, machine learning, random trees, C5.0 decision tree, support vector machine, K-nearest neighbors, discrimination analysis

## Abstract

Substances that do not degrade over time have proven to be harmful to the environment and are dangerous to living organisms. Being able to predict the biodegradability of substances without costly experiments is useful. Recently, the quantitative structure–activity relationship (QSAR) models have proposed effective solutions to this problem. However, the molecular descriptor datasets usually suffer from the problems of unbalanced class distribution, which adversely affects the efficiency and generalization of the derived models. Accordingly, this study aims at validating the performances of balanced random trees (RTs) and boosted C5.0 decision trees (DTs) to construct QSAR models to classify the ready biodegradation of substances and their abilities to deal with unbalanced data. The balanced RTs model algorithm builds individual trees using balanced bootstrap samples, while the boosted C5.0 DT is modeled using cost-sensitive learning. We employed the two-dimensional molecular descriptor dataset, which is publicly available through the University of California, Irvine (UCI) machine learning repository. The molecular descriptors were ranked according to their contributions to the balanced RTs classification process. The performance of the proposed models was compared with previously reported results. Based on the statistical measures, the experimental results showed that the proposed models outperform the classification results of the support vector machine (SVM), K-nearest neighbors (KNN), and discrimination analysis (DA). Classification measures were analyzed in terms of accuracy, sensitivity, specificity, precision, false positive rate, false negative rate, F1 score, receiver operating characteristic (ROC) curve, and area under the ROC curve (AUROC).

## 1. Introduction

The objective of quantitative structure–activity relationship (QSAR) modeling is to discover the relationships between molecular structures and various physical, chemical, and biological activities [1,2]. Computationally, the molecular composition can be described by molecular descriptors that are mathematical representations of chemical information as follows:(1)$$A=f\left(x_{1},x_{2},\ldots ,x_{p}\right)$$
where $\left(x_{1},x_{2},\ldots ,x_{p}\right)$ is a p molecular descriptor vector and A is a certain biological, chemical, or physicochemical activity. Once the QSAR model is developed, it can be used to infer the activity of any new substance from its descriptors without any experimentation [3,4]. The uses of these QSAR models have been extended for environmental purposes, including the ability to test the biodegradability of substances without the need to perform chemical processes. Therefore, several studies have been conducted to apply various machine learning models to predict the biodegradability of chemical compounds (Table 1).

The purpose of this study was to validate the performance of balanced random trees (RTs) and boosted C5.0 decision tree (DTs) models for classifying the biodegradable substances based on two-dimensional molecular descriptors. The RT model is a powerful predictive algorithm for classification and regression purposes with several successful applications [5,6,7]. Random trees and random forest methodologies have the same meaning in the literature. However, the name “random trees” is adopted throughout this paper as it is mentioned in the IBP SPSS Modeler software with the same name. On the other side, the C5.0 model has become one of the most important industry standards for generating decision trees [8,9,10]. The data used in this study represent the 2-D molecular descriptors for a set of ready biodegradable and not-ready biodegradable substances downloaded from the University of California, Irvine (UCI) machine learning repository and donated by Mansouri et al. in [11]. The donors divided the data into three parts: calibration, validation, and external validation subsets. Calibration samples are used to train the predictive models, with one-third of samples representing ready biodegradable (RB) substances, and the other two-thirds samples represent not-ready biodegradable (NRB) ones. Providing such imbalance data directly to the predictive models will lead to undesirable results. This is because the models are designed to increase the overall accuracy, and so, they tend to learn to classify majority samples better than minority samples. Literature has two common approaches to solving this problem: data sampling (oversampling and/or under-sampling) and cost-sensitive learning [12]. In this study, the proposed models followed these two different approaches. The balanced RTs model down-samples the majority class to ensure that the data provided for individual trees are perfectly balanced. On the other side, the boosted C5.0 DTs uses a misclassification cost matrix to modify the biased balance against the minority class. To the best of our knowledge, this is the first time to apply the two models (RTs and boosted C5.0) on the UCI biodegradation data, which adds a novelty to this work. Overall, the contributions of this study are as follows:The data files were obtained from the UCI machine learning repository and grouped into training and test subsets.The descriptive statistics were estimated and the predictive importance was evaluated using mixed type *p*-values based on F-statistics and likelihood ratio chi-squared tests.Machine learning models were proposed for the classification of ready-biodegradable substances using balanced RTs and boosted C5.0 DTs models.Classification performance was evaluated in terms of accuracy, sensitivity, specificity, precision, false positive rate, false negative rate, F1 score, receiver operating characteristic (ROC) curve, and the area under the ROC curve (AUROC).Experimental results show that the proposed models outperform the classification results provided by the support vector machine (SVM), K-nearest neighbors (KNN), and discrimination analysis (DA).The top ten molecular descriptors were ranked according to their contributions to the balanced RTs classification process.

The rest of this paper is prepared as follows: Section 2 presents an overview of the recent developments within the same scope as the current study. Section 3 presents the proposed methodology, including the method pipeline. Section 4 introduces experimental results, model comparisons, and discussion. Finally, conclusions are presented in Section 5.

**Table 1 ijerph-17-09322-t001:** Related work on the classification of biodegradable substances.

Authors	Dataset	Machine Learning Model	Classification Result
Tang et al. [13] 2020	A dataset with 171 compounds was collected from the BIOWIN3 and BIOWIN4 programs and a qualitative dataset from BIOWIN 5 and 6 to validate the reliability of their models.	Four QSAR models were developed for predicting primary and ultimate biodegradation rate rating with multiple linear regression (MLR) and support vector regression (SVR) algorithms.	SVR models have satisfactory goodness-of-fit, robustness, and external predictive abilities.
Lunghini et al. [14] 2020.	Ready biodegradability dataset (2830 compounds), issued by merging several public data sources.	SVM models with linear and RBF kernels, random forest (RF), and Naïve Bayesian (NB) and their Ensemble.	The proposed models showed good performances in terms of predictive power (Balance Accuracy (BA) = 0.74–0.79) and data coverage (83–91%).
Putra et al. [15] 2019	UCI biodegradation dataset.	Artificial neural networks (ANN) and SVM models were built to predict the ready-biodegradation of a chemical compound. Authors reduced the 41 molecular descriptors using principal components analysis (PCA) into four components	SVM achieved 0.77 accuracy, 0.54 sensitivity, and 0.85 specificity. ANN achieved an accuracy of 0.77, sensitivity of 0.61, and specificity of 0.85.
Ballabio et al. [16] 2017.	VEGA, BIOWIN, and Michem datasets, as well as external validation data set	Eight different models: BIOWIN, VEGA, Michem PLS-DA, Michem SVM, Michem KNN, CASE Ultra DTU, Leadscope DTU, and SciQSAR DTU.	VEGA achieved the best non-error rate of 0.88 with a sensitivity of 0.86 and specificity of 0.9.
Zhan et al. [17] 2017	UCI biodegradation dataset.	Naïve Bayes.	AUCs 0.890, 0.921, and 0.901 for training, test and external validation subsets.
Rocha, W. F. C., and Sheen [18] 2016.	UCI biodegradation dataset.	PLS_DA with uncertainty estimation.	They achieved sensitivity of 0.88 and specificity of 0.832 for training data, sensitivity of 0.833, and specificity of 0.87 for test data, and sensitivity of 0.80 and specificity of 0.856 for external validation.
Fernández et al. [19] 2015.	Four Publicly available datasets: BIOWIN, Cheng et al., Japanese MITI, and Lombardo et al.	QSAR models: BIOWIN5, BIOWIN6, START, and VEGA.	BIOWIN6 achieved the best performance with 0.57 MCC, and the consensus of the four models achieved 0.74 MCC
Mansouri et al. [11] 2013.	UCI biodegradation dataset.	KNN, PLS-DA, SVM, consensus 1 and consensus 2 with genetic algorithms.	Best results with consensus 2: training error rate of 0.07, test error rate of 0.09, and external validation error rate of 0.13.
Cheng et al. [20] 2012.	MITI and BIOWIN data sets	SVM, kNN, Naïve Bayes, and C4.5 decision tree using four different feature selections CFS, CART, CHAID, and GA	Best AUCs are 0.856, 0.844, and 0.873 for CART-NB, CHAID-SVM, and GASVM-kNN models, respectively

## 2. Literature Review

Many researchers have been interested in building QSAR models to predict biodegradation information for various compounds. Table 1 presents some of the recent studies in this field.

Tang et al. in [13], proposed four QSAR models to predict the primary and ultimate biodegradation rate using multiple linear regression (MLR) and support vector machine (SVM). The authors collected chemical information about 171. CAS numbers, molecule names, primary and ultimate biodegradation rate rating values were collected from the BIOWIN3 and BIOWIN4 programs. They also collected a qualitative dataset from BIOWIN5 and BIOWIN6 to validate the reliability of their models. Descriptors have been generated using different programs, and models have been implemented in e1071 package of the R language. The authors concluded that SVM models achieved satisfactory goodness-of-fit, robustness, and external predictive ability.

In [14], the authors collected a ready- biodegradability dataset containing 2830 compounds by merging several public data sources. They generated 63 ISIDA molecular descriptor spaces (DS) corresponding to molecular fragments of different sizes, topologies, and coloration. They used this dataset to train SVM with linear and RBF kernels using libSVM, random forest (RF), and Naïve Bayesian (NB) using WEKA software (University of Waikato, Hamilton, New Zealand). These models were externally validated and benchmarked against already-existing tools models. The proposed models showed good performances in terms of predictive power (Balance Accuracy (BA) = 0.74–0.79) and data coverage (83–91%).

In [15,17,18], the authors employed the UCI biodegradation dataset donated by Mansouri et al. in [11]. In [15], the authors reduced the 41 molecular descriptors using principal components analysis (PCA) into four components. They applied ANN and SVM models to predict the ready-biodegradation of a chemical compound. SVM achieved 0.77 accuracy, 0.54 sensitivity, and 0.85 specificity. ANN achieved an accuracy of 0.77, sensitivity of 0.61, and specificity of 0.85. In [17], the author applied the Naïve Bayesian classification (NBC) approach and achieved AUCs 0.890, 0.921, and 0.901 for training, test, and external validation subsets. In [18], the authors applied PLS_DA with uncertainty estimation. They achieved sensitivity of 0.88 and specificity of 0.832 for training data, sensitivity of 0.833, and specificity of 0.87 for test data, and sensitivity of 0.80 and specificity of 0.856 for external validation.

In paper [16], the authors employed VEGA, BIOWIN, and Michem datasets; they applied Eight different models: BIOWIN, VEGA, Michem PLS-DA, Michem SVM, Michem kNN, CASE Ultra DTU, Leadscope DTU, and SciQSAR DTU where the VEGA algorithm achieved the best non-error rate of 0.88 with a sensitivity of 0.86 and specificity 0.9.

In [19], the authors used four publicly available datasets: BIOWIN, Cheng et al. Japanese MITI, and Lombardo et al. datasets, where they applied QSAR models: BIOWIN5, BIOWIN6, START, and VEGA. BIOWIN6 achieved the best performance with 0.57 MCC, and the consensus of the four models achieved 0.74 MCC. MCC stands for Matthews’ correlation coefficient.

Finally, in [20], the MITI and BIOWIN datasets have been used. The authors applied SVM, KNN, Naïve Bayes, and C4.5 decision tree using four different feature selections CFS, CART, CHAID, and GA. The best AUCs are 0.856, 0.844, and 0.873 for CART-NB, CHAID-SVM, and GASVM-KNN models, respectively.

## 3. Method Pipeline

In this study, IBM SPSS modeler was used for experimentation on the UCI biodegradation dataset. The experiments were carried out according to ELTA’s approach to designing business intelligence solutions [21]. ELTA stands for Extract, Load, Transform and Analyze, which outlines the required steps: data collection and preprocessing, feature evaluation and selection, modeling, and finally, performance evaluation and analysis. Figure 1 shows the proposed methodology diagram. The process begins with obtaining the data files from the UCI repository and then defines the data type for every molecular descriptor (continuous and categorical). The type definition is important in decision tree-based modeling (RT and C5.0 models) since their algorithms vary according to the predictor measurements. Then, feature understanding and evaluation steps revolve around assessing the utility of every feature. Features were evaluated according to their predictive importance using *p*-values based on F-statistics (for continuous type) and likelihood ratio chi-square (for the flag and ordinal types). Next, the proposed predictive models (balanced RTs and boosted C5.0 DTs) were adjusted along with other classification models used for comparison purposes: SVM, KNN, and DA models. The performance evaluation and analysis compared the classification results of all these models based on seven different criteria in addition to the analysis using the ROC curve. The molecular descriptors were ranked according to their contributions to the balanced RTs classification process.

### 3.1. Data Collection and Preprocessing

The biodegradation data file can be freely downloaded from the UCI machine learning repository and the publicly available datasets described by Mansouri et al. [11]. Interested reader can inspect the biodegradation dataset by browsing the content of the Excel file enclosed as Appendix A, which includes the molecular descriptors as well as the Chemical Abstracts Service Registry Number (CAS-RN) and Simplified Molecular-Input Line-Entry System (SMILES) code. The datasets present a binary classification problem where each sample is classified as ready biodegradable (RB) or not-ready biodegradable (NRB). The donors divided the data into three parts: calibration subset with 837 samples (284 RB and 553 NRB), validation with 218 samples (72 RB and 146 NRB), and external-validation with 670 samples (479 RB and 191 NRB). They collected these data from three different websites: calibration and validation data are from the webpage of the National Institute of Technology and Evaluation (NITE) of Japan, external validation data are from the work of Chen et al. [20], and the Canadian DSL database (Domestic substances). They used the DRAGON software to extract the two-dimensional molecular descriptors for each compound. Next, they applied the DRAGON filter tool to reduce the number of descriptors and ended up with a list of 781 descriptors. Next, with the help of the genetic algorithm and three different models: SVM, KNN, PLS-DA, the donors reduced the number of descriptors to only 41 descriptors. In this study, the biodegradation data were divided into training and test subsets. The training subset encompassed the original calibration data with 284 RB and 553 NRB samples. While the test subset encompassed the original validation and external validation subsets with 263 RB and 625 NRB samples. Every sample has 41 molecular descriptors described in Table 2. These descriptors are examined to specify their type of measurement (continuous, ordinal, and flagged) since decision tree-based models use different methods depending on the type of input features [22]. However, they all were defined as continuous and were normalized when modeling with other models (SVM, KNN, and DA).

### 3.2. Feature Understanding and Evaluation

Descriptive statistical measures were computed for all continuous, flag, and ordinal features. These measures include minimum, maximum, standard deviation, mode, number of unique values, and predictive importance. The importance was evaluated using *p*-value based on F-statistics for continuous type and likelihood ratio chi-squared for ordinal, and flag ones. These *p*-values are comparable and can be used to evaluate such mixed-type features [23]:
The p-value based on likelihood ratio chi-square $G^{2}$ is calculated by $p\;value=Prob(\chi_{d}^{2}>G^{2})$, where:(2)$$G^{2}=2\sum_{i=1}^{I}\sum_{j=1}^{J}G_{ij}^{2},\;\mathrm{with}\;G_{ij}^{2}=\{\begin{array}{cc} N_{ij}\ln\left(N_{ij}/{\hat{N}}_{ij}\right) & N_{ij>0}\\ 0 & else \end{array}$$
where $N_{ij}=\underset{n}{\sum }f_{n}I\left(x_{n}=i\right)\wedge y_{n}=j$ is the observed cell frequency and ${\hat{N}}_{ij}$ is the expected cell frequency for the cell $\left(x_{n}=i,y_{n}=j\right)$. That is the likelihood ratio chi-square test evaluates the independence between the feature and the output class that involves the difference between the observed and expected frequencies using the contingency table.The p-value based on F statistic is calculated by p value=Prob{F(J−1, N−J)>F}, where:(3)$$F_{i}=\frac{\sum_{j=1}^{J}N_{j}{\left(\mu_{ij}-\mu_{i}\right)}^{2}/\left(J-1\right)}{\sum_{j=1}^{J}\left(N_{j}-1\right)v_{ij}^{2}/\left(N-J\right)}$$
where $\mu_{i}$ is the mean of the *i*th feature, $\mu_{ij}$ is the mean of ith feature in the *j*th class, $v_{ij}^{2}$ is the variance of the *i*th feature in the *j*th class, and $N_{j}$ is the number of samples in the *j*th class. The value of F-statistics increases if the distances between classes are large and the distances within classes are small, which makes the predictive power of the feature is large.

Table 2 shows the resulting importance of all molecular descriptors according to their *p*-values. It is shown that they all have 100% importance except four ones: SdssC, J_Dz(e), Psi_i_A, and Psi_i_1d. They achieve 0.94, 0.89, 0.74, and 0.3 importance, respectively. These values were calculated based on the training subset only. In general, these molecular descriptors represent a good set of features for training machine learning models. Typically, the proposed decision tree-based models (balanced RTs and boosted C5.0 DTs) have their criteria for feature selection [24,25].

### 3.3. Balanced Random Trees (RTs)

The RTs model consists of a set of high-performance decision trees, in which decisions are produced with a majority of votes [23]. To keep low bias and low dependence between trees, the ensemble has two sources of diversity. First, each tree is constructed without pruning on a different training sample (bootstrap). The model allows trees to grow to their maximum—the deeper the tree, the more information it captures about the bootstrap data. Second, at every tree-splitting node, the algorithm randomly picks up a small subset of the available features to choose the best among them to split the data. Given a set of training data, (N samples with p features), the algorithm extract $n_{tree}$ random samples with replacement (bootstrap) to construct $n_{tree}$ individual trees as follows:Step 1:At each decision node, start with all features (p features).Step 2:Select randomly features out of the p available features ($n_{feature}\ll p)$.Step 3:Find the most important feature among the selected $n_{feature}$ features to partition the node data based on a prespecified splitting criterion (Gini’s diversity index).Step 4:Split the current node into two descend nodes (binary partitioning) and continue to develop the tree to its maximum expansion (until there is only one sample in every terminal leaf).Step 5:End and Output $n_{tree}$ trained decision trees (RTs ensemble).

The model classifies a new observation by voting the output of all trees.
(4)$${\hat{f}}_{RTs}\left(x\right)=\arg\max_{y\in \Upsilon }\overset{n_{tree}}{\underset{j=1}{\sum }}I\left(y=h_{j}\left(x\right)\right)$$
where $h_{j}\left(x\right)$ refers to the predicted class label for sample x, by the jth tree, and I(·) is the zero-one loss function.

Typically, each tree is constructed using two-thirds of the training samples (called in-bag data) and validated using the remaining third (called out-of-bag OOB data) (Figure 2) [26]. To address the problem of imbalance in training data, the balanced RTs model constructs individual trees using balanced in-bag data [27]. The model adjusts the sampling rates specific to each target category. In our case, the RB class represents the minority category, and the NRB represents the majority one. The modeling algorithm draws every bootstrap sample from the RB category first and then draws randomly the same number of cases with replacement from the NRB category. The model building process aims to improve the overall classification accuracy and assumes that the data are evenly distributed among the different categories. If the data are not balanced, the classifier will seek to model the majority data more than the minority data, leading to undesirable results. The RTs model has two hyperparameters: the number of trees $n_{tree}$ and the number of randomly selected features at splitting node $n_{feature}$. Typically, $n_{tree}$ is set between 100 and 500 and $n_{feature}$ is set to the square root of the total number of features (p).

#### 3.3.1. Splitting Criteria

The RTs model is an extension of the classification and regression tree (CART) that uses the Gini impurity index to select the splitting feature (and splitting value) with the lowest impurity [27]. At first, the Gini measure evaluates the class label distribution of the data segment in the splitting node. The measure takes values in the range [0, 1], where zero means that all samples in the node belong to the same class. Let the feature x has V values, $x=\left\{x_{1},x_{2},\ldots ,x_{V}\right\}$ at the splitting node t. Partitioning based on x will produce V descendants of the node. Let *N* be the number of samples, $a_{v}$ is the number of samples with value $x_{v}$, $n_{cv}$ is the portion of the *N* samples with value $x_{v}$ belonging to class c, then, the Gini impurity measure is given by:(5)$$I\left(t_{x_{v}}\right)=1-\overset{C}{\underset{c=0}{\sum }}{\left(\frac{n_{cv}}{a_{v}}\right)}^{2}$$

Then, the *Gini index* of the potential splitting feature x is the weighted average of the Gini impurity measure on its different values as follows:(6)$$Gini\;Index\left(t,x\right)=\overset{V}{\underset{v=1}{\sum }}\frac{a_{v}}{N}I\left(t_{x_{v}}\right)$$

Then, the modeling algorithm seeks to split with the lowest Gini Index among the available p features.

#### 3.3.2. Feature Importance

The RTs model has the power to calculate the importance of predictive features [23]. The model directs all the OOB samples down all trees and assessing the predicted output. Then, for every feature ***x***, its values are randomly permuted in the OOB samples, while preserving all other features fixed. Once again, the model produces the predicted outputs (for permuted version). There are two sets of OOB predictions: one set obtained from the real data and one from the permuted version. Then, the importance of a feature x is defined as follows:(7)$$Importance\;\left(x\right)=\frac{1}{n_{tree}}\overset{n_{tree}}{\underset{j}{\sum }}\left(Permuted\;Err_{jx}-Real\;Err_{j}\right)$$

It should be noted, however, that this measure of feature importance only expresses how the RTs model handles training data and cannot be considered as a general measure and is not necessarily valid with other predictive models.

### 3.4. Boosted C5.0 DT Model

The C5.0 DT is a classification tree model developed based on the ID3 and C4.5 decision trees by Ross Quinlan [28,29]. The input features may be continuous or categorical, and the output must be categorical. The DT model can generate simpler and more accurate decision trees with boosting and a bunch of other new techniques with several successful applications [8,9,10,30,31]. In this study, the C5.0 DT model is constructed using cost pruning, sensitive learning, and boosting to tackle the imbalance problem in the biodegradation data.

#### 3.4.1. C5.0 Tree Modeling

The C5.0 tree-building algorithm uses the concepts of entropy, information gain, and gain ratio. Let S be a data segment with N training samples, p features, and C classes, the entropy is a measure of the segment purity as follows:(8)$$Entropy\left(S\right)=-\overset{C}{\underset{i=i}{\sum }}P_{i}\log\left(P_{i}\right)$$
where $P_{i}$ is the portion of S that belongs to the *i*th class. The entropy takes values in the range [0, 1], where zero indicates completely pure data, and one indicates completely random data. Then, for every feature x with V values, the data segment S can be divided into V subsets: $S_{v}$ = {*x* ∈ *S* | *x* = *v*}, and then, the algorithm finds the information gain (IG) of this feature f for segment S as follows:(9)$$IG\left(S,x\right)=Entropy\left(S\right)-\underset{v\in V}{\sum }\frac{\left|S_{v}\right|}{\left|S\right|}Entropy\left(S_{v}\right)$$
where |S| and $\left|S_{v}\right|$ are the number of samples in segments S and $\left|S_{v}\right|$. The smaller the total entropy of the descendant nodes, the more information gained (better classification). To reduce the bias of the information gain, the algorithm computes the Gain Ratio as follows:(10)Gian Ratio(S,x)=Entropy(S)−IG(S,x)

The algorithm continues partitioning the data according to a predetermined minimum number of samples per child branch ($ch_{size}$). Accordingly, the resulting tree has a large number of leaf nodes, which will overfit for noise and anomaly in the training data. To reduce overfitting, the C5.0 algorithm allows the post-pruning process to cut all branches according to a predetermined pruning severity percentage $(p_{Severity}$). That is, the algorithm removes branches with little effect on classification performance. The value of ($p_{Severity}$) controls the confidence level (CF) of the confidence interval $\left[L_{CF},U_{CF}\right]$ [32]. The confidence interval is derived from training data with a confidence level based $P_{Severity}$ as follows:(11)$$CF=1-P_{Severity}/100$$

Lower $p_{Severity}$ prunes fewer branches while higher $p_{Severity}$ prunes more branches. Both $ch_{size}$ and $P_{Severity}$ control the final C5.0 tree structure. For a single C5.0 decision tree, their common values are 2 and 75%, respectively. However, their optimal values can be found using a grid search by experimenting with different values in a suitable range and by assessing the classification loss on the training data. A cross-validation approach may assist in tuning these hyperparameters if the training data are small.

#### 3.4.2. C5.0 Cost-Sensitive Tree

Cost-sensitive learning is a type of learning in data mining that treats various misclassifications differently and aims to reduce the total classification cost [23]. It is one of the important methods in enhancing the ability of classifiers to deal with imbalanced data, as they tend to learn the majority samples with greater accuracy than the minority ones [33]. The method adds costs to data samples (data level) and adjusts the learning algorithm to accept these costs (algorithm level). Typically, the biased balance against the minority class is modified by assuming higher misclassification costs. In the current training data, the proportions of NRB and RB are 0.66 and 0.34, respectively. Therefore, a weighted cost matrix is used to set the misclassification of an RB (as an NRB) to 2 times more costly than the misclassification of an NRB (as an RB). Accordingly, the classifier will pay more attention to the classification of RB samples that are incorrectly predicted.

#### 3.4.3. C5.0 Boosting Algorithm

Boosting is one of the most important improvements in C5.0 compared to the previous C4.5 and ID3. This enhancement approach was based on the research provided by Freund and Schapire, with some special additions to better handle noisy data [34]. It is an augmentation algorithm, which applies a committee of trees and combines their outputs to constitute a powerful ensemble. The trees are constructed successively one by one. The first tree is created as usual. However, the second tree is created, focusing on the wrongly classified samples by the first tree. In addition, the third tree is created, focusing on the wrongly classified samples by the second tree, and so on. Finally, samples are classified by applying the full set of trees using a weighted voting procedure and then augmenting their predictions into one global decision. Specifically, the algorithm uses the training samples to create T boosted trees. Let the $\tau^{t}$ is the constructed tree in the trial t, $\omega_{i}^{t}$ is the weight of the sample i in the trial t, $F_{i}^{t}$ is the normalized factor of $\omega_{i}^{t}$ where their summation equals one, $\overset{N}{\underset{i=1}{\sum }}\;F_{i}^{t}=1$, $\beta_{t}$ is the factor that adjusts weight, and, finally, the indicator function in trial t is defined by.
(12)$$\delta^{t}\left(i\right)=\{\{\begin{array}{c} 1,\;\;if\;the\;i\;sample\;i\;is\;misclassified\\ 0,\;\;if\;the\;i\;sample\;properly\;classified \end{array}$$

Then, the main steps in the C5.0 boosting algorithm are as follows [35]:

**Step** **1:**Initialize the boosting parameters: set the number of trials T (10 is a common value), start iteration t = 1, and initialize weights ωi1=1N (N is the number of training samples)
**Step** **2:**Compute the normalized factors $F_{i}^{t}=\omega_{i}^{t}/\overset{N}{\underset{i=1}{\sum }}\omega_{i}^{t}$**Step** **3:**Assign each $F_{i}^{t}$ to the weight of *i* sample and then generate the tree $\tau^{t}$ according to the current weight distribution.**Step** **4:**Evaluate the error rate of $\tau^{t}$ as $\varepsilon^{t}=\overset{N}{\underset{i=1}{\sum }}F_{i}^{t}\times \delta^{t}\left(i\right)$.**Step** **5:**If $\varepsilon^{t}<0.5$, the trials are closed and set *T* = *t* + 1;else if $\varepsilon^{t}=0$, the trials are terminated and set *T* = *t*;else if $0<\varepsilon^{t}<0.5$ continue to step 6.**Step** **6:**Compute $\beta^{t}=\varepsilon^{t}/\left(1-\varepsilon^{t}\right)$.**Step** **7:**Considering the error rate, adjust the weights using $\beta^{t}$:
$$\omega_{i}^{t+1}=\{\begin{array}{c} \omega_{i}^{t}\times \beta^{t},\;if\;sample\;i\;is\;misclassified\\ \omega_{i}^{t},\;if\;sample\;i\;is\;classified\;correctly \end{array}$$**Step** **8:**if *t* = *T*, the trials are terminated. Else, set *t* = *T* + 1, and go to step 2 to generate the next trial.

Finally, the boosted trees $C^{*}$ are constructed and they work in a committee. Their decision is issued by summing the votes of individual trees $\left(C^{1},C^{2},\ldots ,C^{T}\right)$ weighted by $1/\beta^{t}$ as follows:(13)$$C^{*}=\overset{T}{\underset{t=1}{\sum }}(1/\beta^{t})\times C^{t}$$

A new sample is classified using all trees $C^{t}\left(1\leq t\leq T\right),$ and then, the final vote of each class is counted according to the weight of each tree. The final decision of these boosted trees $C^{*}$ is the class with the highest votes.

## 4. Experimental Results and Discussion

In this paper, we have used the balanced RTs and boosted C5.0 DTs models to build QSAR models for the classification of chemical substances into ready-biodegradable (RB) class or not-ready biodegradable (NRB) class. All preprocessing techniques, models, and evaluation were performed using IBM SPSS modeler version 18 following the ELTA approach. The molecular descriptor dataset was taken from the University of California, Irvine (UCI) machine learning repository. The data donors divided the data into three parts: calibration, validation, and external validation subsets. Calibration samples were used for model training while validation and external validation were used for model testing. The training and testing data are unbalanced as the NRB samples are twice as large as the RB samples. Therefore, it requires special treatment when constructing classification models and also special methods to ensure a fair evaluation. The modeling techniques selected for our experiment were from the decision tree families: the balanced RTs and boosted C5.0 DTs models.

### 4.1. Performance Measures

Typically, the performance evaluation of classification algorithms is performed using statistical measures extracted from the confusion matrix that provides ‘true’ for all properly classified data and ‘false’ for all misclassified data [36]. In this study, seven performance indicators have been used: accuracy, sensitivity, specificity, precision, false positive rate (FPR), false negative rate (FNR), and F1-score. Their definitions and equations can be found in [37]. However, accuracy and F1-score are of particular importance in this study. The accuracy gives a general impression of the efficiency of the model, but it is not sufficient in the case of unbalanced data. On the other hand, the F1-score provides a balance between precision and recall (sensitivity), which is particularly useful when classifying imbalanced data (as in biodegradation data).

### 4.2. Balanced RTs Model

The RTs model has two hyperparameters: the number of trees $n_{tree}$ and the number of randomly selected features at splitting node $n_{feature}$. Typically, $n_{tree}$ is set between 100 and 500 and $n_{feature}$ is set to the square root of the total number of features (p). We set the number of trees to 200 and the randomly selected feature at each splitting node to 7. Increasing the number of trees over 200 does not make a difference, RTs are popular “out-of-the-box” or “off-the-shelf” learning models that enjoy a good predictive performance with somewhat little hyperparameter tuning. All trees were allowed to grow to their maximum extent (minimum samples per child node size to 1). In this study, two RTs models were constructed for comparison: the conventional RTs model constructed using the training data as is and the second the balanced RTs using balanced bootstraps. The performance of the constructed models was very similar when tested again on training data, and both achieved results close to the full score mark. However, the results showed a clear distinction for the balanced model when applying both models to the test data as in Table 3. The sensitivity is improved from 0.75 to 0.80, up 5%, while specificity is worsened from 0.95 to 0.92, down −3%. Likewise, the FNR has fallen by 5%, while the FPR is increased by 3%. While the accuracy is not changed, the table indicates a decrease in precision of 3%, which is less than the improvement in sensitivity resulting in an improvement in the F1 score from 0.8 to 0.81. This indicates that the benefits of the balancing process.

Figure 3 shows the importance of the top 10 molecular descriptors in the balanced RTs model as a bar chart. By referring to Table 2, it is noticed that these top 10 descriptors all have 100% importance based on the *p*-value. This confirms their great importance when classifying biodegradable compounds. In addition, “SpMax_L” is the most important molecular descriptor according to the RTs model, far ahead of the others. It is one of the few features chosen by the three models (SVM, KNN, and DA) in the original paper in [11] (Table 2), which confirms its importance when studying biodegradation substances. Table 4 lists the top ten descriptors, including their definition and their DRAGON block [18]. They belong to five different DRAGON blocks: the 2D matrix-based group (five descriptors), the constitutional indices (two descriptors), the functional group counts (one descriptor), the topological indices (one descriptor), and the 2D atom pairs (one descriptor). SpMax_L, SM6_L, HyWi_B(m), SpPosA_B(p), and SpMax_A all are derived from the molecular graph using different 2D matrices [2]. SpMax_L and SM6_L are derived from the Laplacian matrix, HyWi_B(m) is derived from the Burden matrix weighted by atomic mass (m), SpPosA_B(p) is derived from Burden matrix weighted by atomic polarizability, and SpMax_A is derived from the adjacency matrix. Both C% and nO belong to the constitutional block that takes into account the chemical composition with no information about the overall topology. C% is the percentage of carbon atoms and nO is the number of oxygen atoms. The nCp belongs to the functional groups block that counts the number of atoms/bonds with predicable chemical behavior. The nCp represents the number of terminal primary Carbon atoms with sp3 (the number of connected atoms and lone pairs equal four). The LOC belongs to the topological indices block, which take various structural features into account. The LOC accounts for the pruning partition of the molecular graph. F03[C-N] belongs to the 2D atom pairs block that is based on counting the defined elements of a compound. F03[C-N] counts the frequency of C−N at topological distance 3.

### 4.3. Boosted C5.0 DTs Model

The typical strategy of the C5.0 DT is to build a full tree using a small number of samples per child branch ($ch_{size}$) and then prune this tree to a large extent using a large percentage of pruning severity ($p_{Severity}$) and the common number of boosting trials is 10. These typical values led to a decent performance that is comparable to RTs as shown in Table 5. To address the distortion in the distribution of data between the two categories (RB and NRB), the boosted C5.0 DT can apply a misclassification cost matrix. Therefore, a weighted cost matrix was used to set the misclassification of an RB (as an NRB) to 2 times more costly than the misclassification of an NRB (as an RB). Accordingly, the classifier will give more attention to classifying the RB samples. The training results of the conventional and cost-sensitive models are unaffected, in both cases, the success rate is 100%. However, a clear discrepancy in the test data classification results between the two models as shown in Table 5. The cost-sensitive model causes an improvement in sensitivity from 0.77 to 0.81, with an increase of 4%. The FNR is improved as well as its value decreased from 0.32 m to 0.19. Although this sensitivity (0.81) is better than other models, this improvement comes at the expense of other metrics. The overall accuracy decreases from 0.88 to 0.86, the specificity from 0.93 to 0.88, the precision from 0.82 to 0.73, and the F1-score from 0.79 to 0.77. The FPR increases from 0.07 to 0.12. The improvement in sensitivity (4%) was less than the decrease in precision (5%), which led to a decrease in accuracy and F1-score. The large discrepancy between these training and test results indicates that the boosted C5.0 DT model is more susceptible to the overfitting problem [38].

### 4.4. Model Evaluation and Comparisons

The performances of the proposed models (balanced RTs and boosted C5.0 DTs) are evaluated against three different approaches: SVM, KNN, and DA. These three models are selected for comparison as they were implemented in the original work in [11], where they were configured using genetic algorithms to choose the best features. SVM selected 14 features, KNN selected 12 features and DA selected 23 features, as shown in Table 2. Generally, these three models have several successful bioinformatics applications. According to a recent statistical study on the applications of different artificial intelligence models in the field of healthcare systems, SVM occupied first place while DA came in fourth place, and KNN in eighth place [39]. In this study, we studied the use of these three models once using the features it selected as in Table 2 only and once again using all 41 features. The grid search approach was applied to find the best hyperparameters as follows:(1)SVM model builds a decision hyperplane with the maximal margin width to divide the feature space linearly into two regions (binary classifier) [40]. In the non-linear situations, the model allows a misclassification slake variable ξ around the margin with a regularization constraint C and applies a kernel function to estimate the necessary computations instead of transforming the data into a higher dimensional feature space. In this study, the radial basis function (RBF) kernel function is selected with only one adjustable parameter (σ) [41]. The grid search algorithm is applied to find the optimal values of the regulation parameter C and the RBF sigma σ. The C range was set from 1 to 50, and σ range was set from 0.1 to 10.0. Using all features, we obtained general accuracies of 0.95 and 0.88 for the training and test subsets, respectively, with a formation of C = 1 and σ = 8.7. On the other hand, using only the recommended 14 features as in Table 2, the results were 89.25% and 86.6%, respectively, with C = 1 and σ = 9.1. That is, the SVM accuracies of the training and test subsets are better when using all features. These results are better than those recorded by SVM in the original work in [11] as shown in Table 1.(2)KNN model is based on an intuitive idea that the data samples of the same class should be nearer to each other in the feature space [42]. The model classifies any new sample in three steps. Firstly, the model computes its distance from each of the training samples using a certain distance function. Second, the model selects K number of training samples that have the smallest distances with this new sample. These K samples constitute its neighbors. Finally, the model assigns this new sample to the class to which the largest number of these neighbors belong. Both the number of neighbors K and the distance function play critical roles in the success of this model. In this study, a grid search was applied to find the optimal values of K and the distance function. The K range was set from 3 to 15 while the distance function can be Euclidean or City block distance functions. Using all features, we obtained accuracies of 0.91 and 0.86 for the training and test subsets, respectively, with K = 3 and Euclidean distance. While using only the recommended 12 features, the results were 0.92 and 0.86 with K = 3 and Euclidean distance. In this case, the KNN classification accuracies of the training and testing data are better when using the recommended 12 features and are better than the results in [11] as seen in Table 1.(3)DA model works to find a linear combination of features that distinguish or separate two or more classes of objects. The resulting combination can be used as a linear classifier, or more commonly, to reduce dimensions before subsequent classification [43]. The model works only with continuous input features and categorical outputs. It is assumed that the input features have normal distributions. The violations of the normality assumption are not fatal as long as nonnormality is caused by skewness and not by outliers [23]. If these assumptions are met, DA gives better results. The model requires the determination of methods to compute prior probabilities methods and the covariance matrix. The prior probabilities determine whether the classification coefficients are computed for a priori knowledge of class membership. There are two options: equal probabilities or according to class sizes. The covariance matrix can be estimated within-groups covariance matrix or based on a separate-classes covariance matrix. Using all features, we obtained a classification accuracy of 0.87 and 0.84 for the training and test subsets, respectively. While using only the recommended 23 features, the results were 0.86 and 0.86 for the training and test subsets, respectively. In both experiments, the prior probabilities were computed from the class sizes and the covariance matrix was computed using separate classes. Using the 23 features, the current DA results are similar for the training subset and slightly better for the test subset to those obtained using PLS-DA in [11].

Table 6 presents the results of the test data classification of the five models constructed in this study. The table shows that the balanced RTs model has the best accuracy of 0.89, and F1-score of 81%. It also has the second-best performance depending on sensitivity of 0.8, specificity of 0.93, precision of 0.82, FPR of 0.08, and FNR of 0.2. In all of these measures, in which it is ranked second, it is close to the first. Although the boosted C5.0-based model achieved the best sensitivity of 0.81 and the best FNR of 0.19, however, the rest of its results come close to or less than other systems with an accuracy of 0.86 and F1 scores of 0.77. While the SVM shows a good performance with accuracy of 0.88 and F1-score of 0.79. That is, it is ranked second based on the results of the test data. The KNN achieves accuracy of 0.86 and F1-score of 0.77. Finally, the DA achieves the same accuracy of 0.86 but the F1-score is the least one with 0.75.

### 4.5. Model Evaluation Using (ROC) Curves

The receiver operating characteristic (ROC) graph is an additional key model evaluation procedure [40]. It is a two-dimensional graph of the true positive rate (sensitivity) against the false positive rate (1-specificity). Its importance lies in its ability to compare binary classification models and is not affected by the imbalance in the class distribution. It has been widely used to evaluate diagnostic systems, medical decision-making systems, and machine learning systems [44]. The higher the curve, the higher the sensitivity, and the higher the specificity. As a model with area under the ROC curve (AUROC) that is greater than 0.5, this model works better than a random prediction.

Figure 4 shows the ROC curves and AUROC values of the RB class for the test data, for all the studied classification models. It was clear that the balanced RTs and the enhanced C5.0 DTs came in first and second places with AUROC of 0.92 and 0.91, respectively. On the other hand, SVM, KNN, and DA achieved only 0.88. 0.88, and 0.90 AUROC, respectively. That is, the two proposed models outperform the other three models based on the AUROC results. This demonstrates their ability to efficiently handle biodegradation data. Compared to the findings reported in the preceding research in [11,15,17,18], the results of the proposed models are better for both training and test subsets. The results highlight the ability of these two tree-based ensembles to accommodate different feature types and to model such QSAR nonlinear relationships. In addition, the need remains to use unbalanced data handling methods that lead to better results.

## 5. Conclusions

The quantitative structure–activity relationship (QSAR) models provide effective solutions for classifying the degradation activity of different materials without resorting to chemical experiments. This study presented two different models: balanced RTs and boosted C5.0 DTs to build QSAR models for classifying degradable materials. Both models were designed to achieve the highest accuracy in predicting biodegradable materials according to ELTA approach. Biodegradation data files were collected and samples were grouped into training and test subsets. These samples fall into two classes ready biodegradable (RB) and not-ready biodegradable (NRB) molecules. The NRB samples are twice the RB ones in the training subset. The input molecular descriptors were explored and their measurement type, descriptive statistics, and predictive importance were identified. Their predictive importance was evaluated using the *p*-value based on the F-statistics for continuous type and the likelihood ratio chi-square for ordinal, and flag types. The two models were designed considering the imbalance in class distribution: the balanced RTs model algorithm down-sampled the majority class to build individual trees using perfectly balanced bootstrap samples; while the boosted C5.0 DTs is modeled using cost-sensitive learning. To measure the efficiency of the two proposed models, seven performance measures were used: accuracy, sensitivity, specificity, precision, false positive rate, false negative rate, and F1 score. Their performances were compared with three of the latest machine learning models: SVM, KNN, and DA. Additional comparisons were also made based on the ROC curves and the areas below them. The accuracy, F1-score, and AUROC results of the testing data demonstrated the advantage of the balanced RTs model and its ability to handle biodegradation data with an accuracy of 0.89, F1-score of 0.81, and AUROC of 0.92 more than the results reported in the preceding research that used SVM, KNN, and DA models. The boosted C5.0 provided comparable results (AUROC of 0.91) with the best sensitivity test result (0.81). The top ten molecular descriptors were ranked according to their contributions to the balanced RTs classification process with five belonging to the 2D matrix-based group. The “SpMax_L”, leading eigenvalue derived from the Laplace matrix, proved to be the most important feature far ahead of the others.

## Figures and Tables

**Figure 1 ijerph-17-09322-f001:**
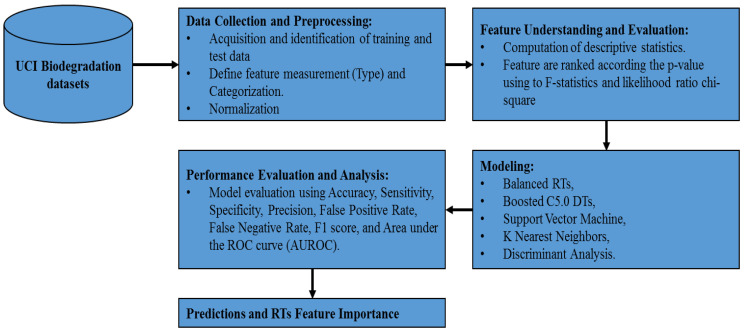
The proposed methodology based on the Extract, Load, Transform and Analyze (ELTA) approach [21].

**Figure 2 ijerph-17-09322-f002:**
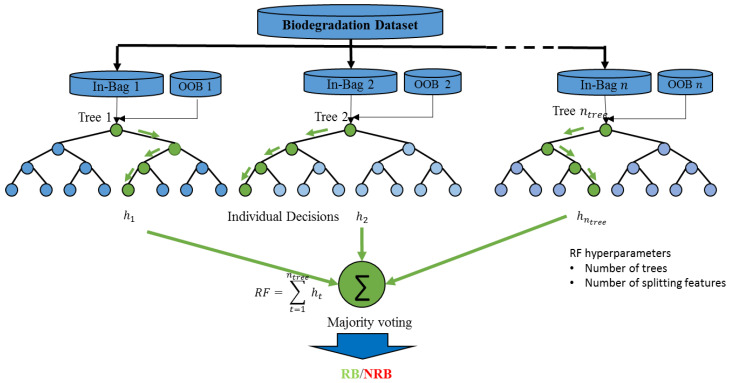
Structure of random forest.

**Figure 3 ijerph-17-09322-f003:**
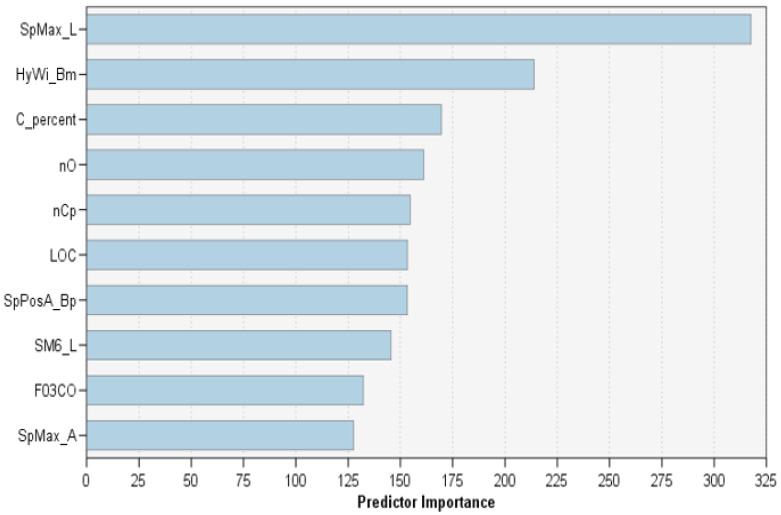
The ten most important predictive molecular descriptors according to the balanced RTs model. They are SpMax_L, HyWi_B(m), C%, nO, nCp, LOC, SpPosA_B(p), SM6_L, F03[C-O], and SpMax_A. Their definitions and DRAGON blocks are shown in Table 4.

**Figure 4 ijerph-17-09322-f004:**
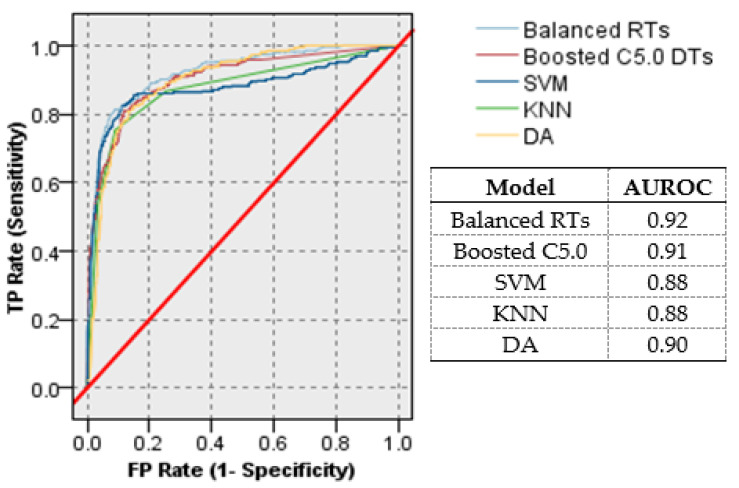
ROC curves and the area under the ROC curve (AUROC) values of the RB class for the test data. The higher the curve to the left, the higher the model’s predictive power than the others.

**Table 2 ijerph-17-09322-t002:** Description of the features used for the classification of biodegradable chemicals with their type, minimum, maximum, mean, standard deviation, mode, number of unique values, and predictive importance.

Descriptor	Definition	Selecting Model	Type	Min	Max	Mean	Std. Dev ^1^	Mode ^2^	Unique ^3^	Importance
B01[C-Br]	Presence/absence of C−Br at topological distance 1	PLS-DA	Flag	0	1	-	-	0	2	1.0
B03[C-Cl]	Presence/absence of C−Cl at topological distance 3	PLS-DA	Flag	0	1	-	-	0	2	1.0
B04[C-Br]	Presence/absence of C−Br at topological distance 4	PLS-DA	Flag	0	1	-	-	0	2	1.0
C%	Percentage of C atoms	KNN,PLS-DA	Continuous	0.0	62.5	37.5	9.1	33.3	-	1.0
C-026	R−CX−R	SVM	Ordinal	0	12	-	-	0	12	1.0
F01[N-N]	Frequency of N−N at topological distance 1	KNN	Ordinal	0	3	-	-	0	4	1.0
F02[C-N]	Frequency of C−N at topological distance 2	SVM	Ordinal	0	24	-	-	0	22	1.0
F03[C-N]	Frequency of C−N at topological distance 3	KNN	Ordinal	0	44	-	-	0	28	1.0
F03[C-O]	Frequency of C−O at topological distance 3	PLS-DA	Ordinal	0	42	-	-	0	33	1.0
F04CN	F04[C-N] frequency of C−N at topological distance 4 2D atom pairs	KNN,PLS-DA	Ordinal	0	36	-	-	0	25	1.0
HyWi_B(m)	Hyper-Wiener-like index (log function) from Burden matrix weighted by mass	PLS-DA	Continuous	1.5	6.3	3.6	0.6	3.6	-	1.0
J_Dz(e)	Balaban-like index from Barysz matrix weighted by Sanderson electronegativity	KNN	Continuous	0.8	9.2	3.0	0.9	2.0	-	0.9
LOC	Lopping centric index	PLS-DA	Continuous	0.0	4.5	1.4	0.8	0.0	-	1.0
Me	Mean atomic Sanderson electronegativity (scaled on Carbon atom)	PLS-DA	Continuous	1.0	1.3	1.0	0.0	1.0	-	1.0
Mi	Mean first ionization potential (scaled on carbon atom)	PLS-DA	Continuous	1.0	1.4	1.1	0.0	1.13	-	1.0
N-073	Ar2NH/Ar3N/Ar2N−Al/R···N···R	PLS-DA	Ordinal	0	3	-	-	0.0	4	1.0
nArCOOR	Number of esters (aromatic)	SVM	Ordinal	0	4	-	-	0.0	5	1.0
nArNO2	Number of nitro groups (aromatic)	PLS-DA	Ordinal	0	4	-	-	0.0	5	1.0
nCb-	Number of substituted benzene C(sp2)	KNN,SVM	Ordinal	0	18	-	-	0.0	16	1.0
nCIR	Number of circuits	PLS-DA	Ordinal	0	147	-	-	1.0	22	1.0
nCp	Number of terminal primary C(sp3)	KNN	Ordinal	0	24	-	-	0.0	15	1.0
nCrt	Number of ring tertiary C(sp3)	SVM	Ordinal	0	8	-	-	0.0	9	1.0
nCRX3	Number of CRX3	PLS-DA	Ordinal	0	3	-	-	0	4	1.0
nHDon	Number of donor atoms for H-bonds (N and O)	SVM	Ordinal	0	16	-	-	0	11	1.0
nHM	Number of heavy atoms	KNN	Ordinal	0	12	-	-	0	11	1.0
nN	Number of nitrogen atoms	SVM	Ordinal	0	10	-	-	0	11	1.0
nN-N	Number of N hydrazines	PLS-DA,SVM	Ordinal	0	2	-	-	0	3	1.0
nO	Number of oxygen atoms	KNN,PLS-DA	Ordinal	0	18	-	-	0	15	1.0
NssssC	Number of atoms of type ssssC atom-type	KNN,SVM	Ordinal	0	16	-	-	0	14	1.0
nX	Number of halogen atoms	SVM	Ordinal	0	33	-	-	0	20	1.0
Psi_i_1d	Intrinsic state pseudoconnectivity index−type 1d	PLS-DA	Continuous	1−.1	1.1	0.0	0.2	0.0	-	0.3
Psi_i_A	Intrinsic state pseudoconnectivity index--type S average	SVM	Continuous	1.5	5.8	2.5	0.7	2.8	-	0.7
SdO	Sum of dO E-states	PLS-DA	Continuous	0.0	95.1	10.3	13.8	0.0	-	1.0
SdssC	Sum of dssC E-states	KNN	Continuous	7−.6	4.7	0−.2	0.8	0.0	-	0.9
SM6_B(m)	Spectral moment of order 6 from Burden matrix weighted by mass	SVM	Continuous	4.7	17.2	8.8	1.4	8.6	-	1.0
SM6_L	Spectral moment of order 6 from Laplace matrix	PLS-DA	Continuous	4.2	12.7	10.1	1.0	8.6	-	1.0
SpMax_A	Leading eigenvalue from adjacency matrix (Lovasz−Pelikan index)	PLS-DA	Continuous	1.0	2.9	2.2	0.2	2.0	-	1.0
SpMax_B(m)	Leading eigenvalue from Burden matrix weighted by mass	SVM	Continuous	2.2	17.6	4.0	1.2	6.9	-	1.0
SpMax_L	Leading eigenvalue from Laplace matrix	KNN,PLS-DA,SVM	Continuous	2.0	6.5	4.8	0.6	4.7	-	1.0
SpPosA_B(p)	Normalized spectral positive sum from Burden matrix weighted by polarizability	PLS-DA	Continuous	0.9	1.6	1.2	0.1	1.3	-	1.0
TI2_L	Second Mohar index from Laplace matrix	PLS-DA	Continuous	0.4	17.8	3.0	2.3	1.5	-	1.0

^1^ Std. Dev. indicates the standard deviation of feature values. ^2^ Mode refers is the value that appears most often in feature values. ^3^ Unique counts the distinct values appear in feature values.

**Table 3 ijerph-17-09322-t003:** Experimental results of conventional random trees (RTs) and balanced RTs classification models for the test subset.

Model	Sensitivity	Specificity	Accuracy	Precision	F1-Score	FPR	FNR
RTs	0.75	0.95	0.89	0.86	0.80	0.05	0.25
Balanced RTs	0.80	0.92	0.89	0.82	0.81	0.08	0.20

**Table 4 ijerph-17-09322-t004:** The definition and DRAGON block of the top 10 molecular descriptors ranked according to their contribution to the balanced RTs classification process.

Molecular Descriptor	Definition	DRAGON Block
SpMax_L	leading eigenvalue from Laplace matrix	2D matrix-based
HyWi_B(m)	hyper-Wiener-like index (log function) from Burden matrix weighted by mass	2D matrix-based
C%	percentage of C atoms	constitutional indices
nO	number of oxygen atoms	constitutional indices
nCp	number of terminal primary C(sp3)	functional group counts
LOC	lopping centric index	topological indices
SpPosA_B(p)	normalized spectral positive sum from Burden matrix weighted by polarizability	2D matrix-based
SM6_L	spectral moment of order 6 from Laplace matrix	2D matrix-based
F03[C-N]	frequency of C−N at topological distance 3	2D atom pairs
SpMax_A	leading eigenvalue from adjacency matrix (Lovasz−Pelikan index)	2D matrix-based

**Table 5 ijerph-17-09322-t005:** Experimental results of boosted C5.0 decision trees (DTs) without and with misclassification cost matrix for the test subset.

Model	Sensitivity	Specificity	Accuracy	Precision	F1-Score	FPR	FNR
Boosted C5.0 DTs	0.77	0.93	0.88	0.82	0.79	0.07	0.23
Boosted C5.0 DTs with misclassification cost matrix	0.81	0.88	0.86	0.73	0.77	0.12	0.19

**Table 6 ijerph-17-09322-t006:** Statistical results of individual classification models on the test subset of 888 substances (263 RB and 625 NRB).

Model	Sensitivity	Specificity	Accuracy	Precision	F1-Score	FPR	FNR
Balanced RTs	0.80	0.93	0.89	0.82	0.81	0.08	0.20
Boosted C5.0 DTs with misclassification cost matrix	0.81	0.88	0.86	0.73	0.77	0.13	0.19
SVM	0.73	0.94	0.88	0.85	0.79	0.06	0.27
KNN	0.75	0.91	0.86	0.78	0.77	0.09	0.25
DA	0.73	0.91	0.86	0.77	0.75	0.09	0.27

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
