# Peer review of "Classification of Biodegradable Substances Using Balanced Random Trees and Boosted C5.0 Decision Trees"

_ijerph, 2020, doi:10.3390/ijerph17249322_

Round 1

Reviewer 1 Report

This paper presents an analysis of the C5 boosted decision tree algorithm as applied to the UCI ready-biodegradability dataset. In its current form, the paper does not present any particularly novel conclusions and many changes are needed to make it publishable.

The first major issue is that the paper is 23 pages long, and most of the pages are devoted to concepts that do not need to be rehashed in this paper. Such concepts include explaining how a QSAR model works, defining the performance metrics for a classification model such as sensitivity and specificity, explaining a confusion matrix, and explaining how decision trees perform their splits. Many of these concepts are well-described in other works to which the authors could simply refer. This is especially important when explaining the C5 tree algorithm; this algorithm is of central importance to the paper, but it is also well-established in the literature and the authors should not be spending words describing it in meticulous detail. Removing this extraneous explanation would free up room in the paper, and bandwidth in the mind of the reader, to focus on the important conclusions.

Second, the impact of the paper's results is not clear. The major results seem to be in Tables 5, 7, and 8, and in Figure 5. I ignore the training performance in Table 6 because sens and spec of 1.0 is not terribly surprising performance for a decision tree on its own training set. As for the test set, sensitivities and specificities of 0.8-0.9 are nothing to write home about. For those performance metrics where the C5 algorithm performs better, the authors do not discuss whether this is actually statistically better or merely a fluke. Furthermore, the C5 algorithm performs worse than the other ML models for many metrics. For example, every ML model has a higher specificity than the C5 trees, as seen in Table 7. The discussion of Table 7 passes over this observation in utter silence.

In conclusion, this paper spends many pages on background topics it does not need to, and needs to have a much more forthright and complete discussion of its central results.

Author Response

All authors are grateful and greatly appreciated the reviewers' efforts and time.

We made a few adjustments and wrote them in red in the revised version.
We have attached a file to respond to all of your comments

Reviewer 2 Report

This paper builds balanced categorical models of "biodegradability" using standard QSAR methods and a public domain database. Many journals would not consider applying off-the-shelf methods against a public domain database sufficiently novel to warrant publication unless some new insight is gained. The authors themselves point out several similar models in the literature. I will leave that up to the editors, but the authors should clearly state why their contribution is a signficant contribution.

Scientific issues:

Some description of what kind of compounds are being modeled is needed. Are they "chemicals", drug-like, etc. Distribution of molecular weight and ALOGP of the training/test sets should be shown. Supporting material should be provided with the structures (e.g. in SMILES format) of the compounds in this study.

These training sets are not particularly unbalanced, e.g. 1:2, whereas in my experience difficulties occur with ratios < 1:5. Balanced treatment cannot hurt however, so I would accept the procedure.

Usually the accuracy of a model fitting its own training set depends very sensistively on the nature of the algorithm, and thus it is meaningless to compare methods that way. Only predicting the "test set" is meaningful. Thus the "training set fits" should be omitted from all tables and Figures.

Do the other methods (SVM, KNN, etc.) do worse because they do not have an explicit way of "balancing" the training set? Although, as stated above, this dataset is not particularly unbalanced in the sense usually used.

Some useful chemical interpetation of the important descriptors in Figure 4 would be useful.

Editorial issues:

Most of this paper consists of descriptions of how various QSAR methods work. It is not necessary to describe them in detail, since they are well-described in other publications. A one or two sentence summary would be sufficient.

What the authors call "random trees" seems to be what most people would call "random forest".  There is a third parameter in random forest, which is the number of molecules in a node below which the node will not be split further. For classification models this is usually set to 1, so I guess it is ok to omit it.

Table 1. Need the year on Tang et al.

Table 2. I don't understand Table 2. Is it a metric of predictive importances of single features estimated before model building. Why is this here since Figure 4 gives a more relevent determinations of the usefulness of descriptors to the model? BTW the "permutation" approach to estimating descriptor importances is general and not specific to recursive partitioning methods like random forest.

The sentence in the abstract should be more like:

Substances that do not degrade over time have proven can be harmful to the environment and and is dangerous to living organisms. Being able to predict the biodegradability of substances without costly experiments is useful.

Author Response

(The authors gave the same response as above.)

Reviewer 3 Report

Journal: IJERPH (ISSN 1660-4601)

Manuscript ID: ijerph-986986

Title: Classification of Biodegradable Substances Using Balanced Random Trees and Boosted C5.0 Decision Trees

Overall  Comments and Suggestions for Authors

Point. 1

For classification of biodegradable substances, this manuscript has proposed suitably with balanced random forest and boosted decision trees. Overall materials, selected algorithm, and machine learning process were considered to be reasonable.

Point. 2

“4.1. Performance measures” is not a part of results and discussion. This should be also considered as Methods section instead of Results. Authors need to revise this context.

Point. 3

Regarding Figure 6, it would be better for authors to discuss more about the interpretation of Importance Variables.

Point. 4

In addition to model evaluation using (ROC) curves, it would be more desirable and convincing to describe about the main model performance and predictor selections by comparing to the possible preceding researches.

Point. 5

The overall analysis was performed well following the methodology, but it seems that the relevant interpretation was a bit missing. If it can be explained additionally, this manuscript would be more persuasive.

Author Response

(The authors gave the same response as above.)

Round 2

Reviewer 2 Report

This paper builds balanced categorical models of "biodegradability" using standard QSAR methods and a public domain database.

The authors fixed up some typos and, importantly, added supplementary material containing the SMILES strings of the compounds in the dataset. I can now tell what the size and lipophilicity of the molecules are. On the average they are are smaller and more lipophilic than drugs. This also allowed me to run my own QSAR modeling on the data to help me interpret what molecular features are important.

I am still having the following issues:

The fact that a particular combinations of methods were tried in this paper but not previously tried elsewhere, is not enough to declare this paper novel, in my opinion. Again, I will leave that to the editors.

Scientific issues:

I still think it is not meaningful to discuss "goodness-of-fit" metrics for the training set. This will always look good in random forest, relative to some of the other methods. Only predicting the "test set" is meaningful. Thus the "training set" should be omitted from all tables and Figures. It is fairly well understood that random forest will do better than methods such as KNN and DA for most datasets.

I think the "eigenvalue" descriptors in Figure 3 and Table 4 are unintepretable in terms of something a chemist could make a decision on. I redid the QSAR using substructure-type descriptors (e.g. atom pairs, ECFP4). Although these were not quite as predictive as computable physical properties descriptors, they do give some insight. For example, aliphatic hydrocarbon chains (-CH2-CH2-CH2-) seem to common among the RB molecules, aromatic rings and basic amines seem to be more common among the NRB molecules. Interpretations of that type would probably make this paper more "novel."

Editorial issues:

Some of the descriptions of how QSAR methods work have been shortened a little, but I still think a one or two sentence summary with a reference would be sufficient in most cases.

I accept "random trees" instead of "random forest" as the name of the method, but "random forest" is the more common term in the literature.

Given that we have Figure 3, which gives a metric of importances of descriptors to the model, why do we need Table 2 which gives a probability that a descriptor is correlated with the activity. This seems less relevant.

Author Response

Reviewer 2

All authors are grateful and highly appreciating the reviewers' effort and time.

Reviewer’s Comment

The authors fixed up some typos and, importantly, added supplementary material containing the SMILES strings of the compounds in the dataset. I can now tell what the size and lipophilicity of the molecules are. On the average, they are smaller and more lipophilic than drugs. This also allowed me to run my own QSAR modeling on the data to help me interpret what molecular features are important.

Authors Reply

The authors would like to thank the reviewer for these positive words. At the same time, we are very delighted to know that this dataset allowed the reviewer to run his own QSAR modeling.

I am still having the following issues:

Reviewer’s Comment

The fact that a particular combinations of methods were tried in this paper but not previously tried elsewhere, is not enough to declare this paper novel, in my opinion. Again, I will leave that to the editors.

Authors' Reply

The authors appreciate and respect the reviewer’s viewpoint regarding the novelty of the paper.

From another viewpoint, the authors think that if a particular combination of methods were tried in this paper but not previously tried elsewhere, this declares that the methodology is “new” which indicates its ‘novelty’, otherwise it is a repeated work.

Scientific issues:

Reviewer’s Comment

 I still think it is not meaningful to discuss "goodness-of-fit" metrics for the training set. This will always look good in random forest, relative to some of the other methods. Only predicting the "test set" is meaningful. Thus the "training set" should be omitted from all tables and Figures. It is fairly well understood that random forest will do better than methods such as KNN and DA for most datasets.

Authors' Reply

The authors would like to thanks the reviewer for this point. The "training set" has been omitted from all tables and Figures as suggested by the reviewer.

Reviewer’s Comment

I think the "eigenvalue" descriptors in Figure 3 and Table 4 are unintepretable in terms of something a chemist could make a decision on. I redid the QSAR using substructure-type descriptors (e.g. atom pairs, ECFP4). Although these were not quite as predictive as computable physical properties descriptors, they do give some insight. For example, aliphatic hydrocarbon chains (-CH2-CH2-CH2-) seem to common among the RB molecules, aromatic rings and basic amines seem to be more common among the NRB molecules. Interpretations of that type would probably make this paper more "novel."

Authors' Reply

The authors would like to thanks the reviewer for this point. This study is about the QSAR modelling based on molecular descriptors. Typically, these descriptors are derived by applying principles from several different theories (connectivity, adjacency, topology, walk path, information,…).

To respond to this comment, we have added the following sentences to Section 4.2. to shed more light on the ten most important descriptors and how they are calculated from the molecule structure, graph, topology,…

Table 4 lists the top ten descriptors, including their definition and their DRAGON block [18]. They belong to five different DRAGON blocks: the 2D matrix-based group (five descriptors), the constitutional indices (two descriptors), the functional group counts (one descriptor), the topological indices (one descriptor), and the 2D atom pairs (one descriptor). SpMax_L, SM6_L, HyWi_B (m), SpPosA_B (p), and SpMax_A all are derived from the molecular graph using different 2D matrices [2]. SpMax_L and SM6_L are derived from the Laplacian matrix, HyWi_B (m) is derived from the Burden matrix weighted by atomic mass (m), SpPosA_B (p) is derived from Burden matrix weighted by atomic polarizability, and SpMax_A is derived from the adjacency matrix. Both C% and nO belong to the constitutional block that takes into account the chemical composition with no information about the overall topology. C% is the percentage of carbon atoms and nO is the number of oxygen atoms. The nCp belongs to the functional groups block that counts the number of atoms/bonds with predicable chemical behavior. The nCp represents the number of terminal primary Carbon atoms with sp3 (the number of connected atoms and lone pairs equal four). The LOC belongs to the topological indices block, which take various structural features into account. The LOC accounts for the pruning partition of the molecular graph. F03[C-N] belongs to the 2D atom pairs block that is based on counting the defined elements of a compound. F03[C-N] counts the frequency of C−N at topological distance 3.

Editorial issues:

Reviewer’s Comment

 Some of the descriptions of how QSAR methods work have been shortened a little, but I still think a one or two sentence summary with a reference would be sufficient in most cases.

Authors Reply

The authors thank the reviewer for this comment and the QSAR methods have been shortened more in the introduction section.

Reviewer’s Comment

 I accept "random trees" instead of "random forest" as the name of the method, but "random forest" is the more common term in the literature.

Authors' Reply

The authors would like to thank the reviewer to improve the readability of this work. Therefore, we have added the following statement to the manuscript:

Random trees and random forest methodologies have the same meaning in the literature. However, the name “random trees” is adopted throughout this paper as it is mentioned in the IBP SPSS Modeler software with the same name.

Reviewer’s Comment

Given that we have Figure 3, which gives a metric of importances of descriptors to the model, why do we need Table 2 which gives a probability that a descriptor is correlated with the activity. This seems less relevant.

Authors' Reply

The authors thank the reviewer for this comment.

Some references indicate that the feature ranking by the RTs model only expresses how that model deals with training data and cannot be considered a general metric.

Table 2 provides the descriptive statistics on all molecular descriptors used, including the predictive importance depending on the mixed type p-value. In the literature, there are three common feature importance methodologies: filter, wrapper, and embedded. Each one has its own advantages. We referred to a recent survey on feature selection in reference [25]. The relevancy of Table 2 is explained in the following:

·         The original paper (reference [11]) generated this dataset using the wrapper methods with SVM, KNN, and DA, as described in Section 4.4, and Table 2.

·         SVM selected 14 features, KNN selected 12 features and DA selected 23 features.

·         In this paper, features are ranked using mixed type p-value (filter method) as shown in Table 2 and using random trees (embedded method) as shown in Figure 3 and Table 4. In all of these methods.

·         The ten most important features of RTs model have full filter significance with a p-value and were selected previously by other predictive models.
